# Validation of the Glasgow Antipsychotic Side-Effect Scale (GASS) in an Italian Sample of Patients with Stable Schizophrenia and Bipolar Spectrum Disorders

**DOI:** 10.3390/brainsci12070891

**Published:** 2022-07-07

**Authors:** Alessandro Rodolico, Carmen Concerto, Alessia Ciancio, Spyridon Siafis, Laura Fusar-Poli, Carla Benedicta Romano, Elisa Vita Scavo, Antonino Petralia, Salvatore Salomone, Maria Salvina Signorelli, Stefan Leucht, Eugenio Aguglia

**Affiliations:** 1Department of Clinical and Experimental Medicine, Institute of Psychiatry, University of Catania, 95123 Catania, Italy; c.concerto@policlinico.unict.it (C.C.); alessia.ciancio@gmail.com (A.C.); laura.fusarpoli@unict.it (L.F.-P.); romanocarlab@gmail.com (C.B.R.); elisavita.scavo@tiscali.it (E.V.S.); petralia@unict.it (A.P.); maria.signorelli@unict.it (M.S.S.); eugenio.aguglia@unict.it (E.A.); 2Department of Psychiatry and Psychotherapy, School of Medicine, Technical University of Munich, 81675 Munich, Germany; spyridon.siafis@tum.de (S.S.); stefan.leucht@tum.de (S.L.); 3Department of Biomedical and Biotechnological Sciences, School of Medicine, University of Catania, 94123 Catania, Italy

**Keywords:** antipsychotics, side-effects, scale, validation, sensitivity, specificity, functioning, quality of life, schizophrenia, bipolar disorder

## Abstract

Antipsychotics are a class of psychotropic drugs that improve psychotic symptoms and reduce relapse risk. However, they may cause side effects (SE) that impact patients’ quality of life and psychosocial functioning. Therefore, there is a need for practical tools to identify them and possibly intervene. The objective of the present study was to translate into Italian the Glasgow Antipsychotic Side Effect Scale (GASS), which is suggested as the questionnaire of choice to collect SE reported by patients treated with antipsychotics. We administered the GASS and the Udvalg for Kliniske Undersøgelser (UKU) SE scale—which is considered the gold standard—to 100 stable patients with schizophrenia and bipolar spectrum disorders. We measured the structural validity, internal consistency, concurrent criterion validity, construct validity, and clinical feasibility. GASS was characterized by modest structural validity and good internal consistency. The binary correlations concerning the presence of specific symptoms investigated with the GASS and the UKU were strong or relatively strong for only half of them. The GASS total scale score was inversely related to patients’ quality of life and psychosocial functioning. The GASS is useful to briefly assess the burden of antipsychotic SE (~5 min) but is not optimal in identifying them.

## 1. Introduction

Antipsychotic (AP) drugs are widely used for the treatment of several psychiatric conditions, including schizophrenia and bipolar disorder [1,2,3], which require long-term specific treatment to manage symptom severity, improve outcomes, and reduce relapses [4].

Both first- and second-generation antipsychotics may cause a wide range of side effects (SE)—including weight gain, sedation, prolactin-related sexual dysfunction, cardiovascular problems, and extrapyramidal symptoms to a variable extent [5]—which may impact patients’ quality of life and psychosocial functioning [6].

Guidelines suggest constant monitoring of SE to ensure good treatment efficacy without losing sight of tolerability [7]. Since the 1970s, several scales have been developed to evaluate the SE induced by AP treatment [8,9,10]. Some of them evaluate specific SE such as extrapyramidal and sexual SE [9,11], while others are more extensive and consider various SE categories [10,12]. Hetero-administered and self-administered scales have been broadly used [9,10,12,13]. Among the former, the Udvalg for Kliniske Undersøgelser (UKU) SE rating scale for clinicians [12] is the most widely used in the research field [14]. UKU is usually administered by trained health professionals via an extensive semi-structured interview. It might be considered a difficult tool to use in daily clinical practice owing to the fact that it is time-consuming. Indeed, due to this limitation, a recent systematic review recommended using brief, self-report, multi-domain questionnaires to screen for SE in clinical practice [14].

The Glasgow Antipsychotic Rating Scale (GASS) was suggested by a consensus of over 100 professionals and users as the standard patient-reported outcome measure used to collect data about AP-SE [15]. The GASS scale selection process was based on a sequence of steps. Initially, two patient focus groups selected the most important outcomes for them. Subsequently, the stakeholders chose only the “essential” outcomes. Next, they assessed them with the Consensus-based Standards for the Selection of Health Outcomes Measurement Instruments (COSMIN) checklist to check the psychometric properties of measures, including reliability, validity, and responsiveness [16]. Finally, the GASS was chosen to monitor treatment side effects. It consists of a set of 22 self-explanatory questions expressed in easy-to-understand English. The patient indicates the frequency and the distress determined by the specific SE. The higher the sum of the frequencies of the SEs, the higher the burden the SEs determine. Its brevity permits the completion of the patient self-assessment in about 5 minutes [17]. The GASS was originally validated against another self-report scale, the Liverpool University Neuroleptic Side-effect Rating Scale (LUNSERS) [10], showing good discriminatory power, construct validity, and test-retest reliability. These advantages have been confirmed by Bock et al. [18] with a direct validation against the UKU.

The present work aimed to translate the GASS into Italian, according to standard practices, and to measure its structural validity, internal consistency, concurrent criterion validity (against the UKU scale), and clinical feasibility.

## 2. Materials and Methods

### 2.1. Participants

The participants were inpatients and outpatients recruited from the Psychiatry Unit of the University of Catania, Catania, Italy. Inclusion criteria were: (a) age ≥ 18 years; (b) being an inpatient or outpatient; (c) diagnosis of schizophrenia spectrum disorder or bipolar spectrum disorder based on DSM-5 criteria; (d) having been on treatment with at least one AP for at least 6 months (persistent use of the same AP was not required); (e) absence of positive symptoms at the time of recruitment (defined with a score ≤3 on the Positive and Negative Syndrome Scale [PANSS] p1-delusion, p3-hallucinatory behavior, g9-unusual thought content inspired by Andreasen’s remission criteria for positive symptoms); (f) absence of depressive or manic symptoms at the time of recruitment (defined as a score < 10 on the Montgomery–Asberg Depression Rating Scale [MADRS] and <7 on the Young Mania Rating Scale [YMRS]); (g) presence of good insight (PANSS g12-lack of judgment & insight ≤3); (h) absence of delusions and hallucination for bipolar patients; (i) sufficient understanding of the proposed questionnaires; and (j) ability to read and understand the informed consent documentation. 

Patients were excluded if: (a) they were treated in the context of a compulsory intervention; (b) they presented concomitant organic diseases; (c) they declared they were currently using psychoactive substances; (d) they presented other neurological conditions (i.e., epilepsy, movement disorders, intellectual disability, dementia, etc.); or (e) they presented any condition that would prevent the completion of the assessment. The following demographic and clinical data were collected: age, sex, education, marital status, employment status, smoking status, concomitant pathologies, illness-related data (illness duration, hospitalizations, actual recruitment setting), and drug-related data (antipsychotic used, olanzapine oral-equivalents, administration route, concomitant psychotropic medications).

### 2.2. Instruments

The GASS is a self-administered scale initially developed in English [17] and translated into other languages [18,19,20]. It is a 22-item self-rated questionnaire used to assess AP-induced weight gain; sedation; central nervous system (CNS), cardiovascular, gastrointestinal, and genitourinary functioning; extrapyramidal and anticholinergic activity; diabetes; and prolactin-related SE. For each item, patients can indicate the frequency of the reported SE (Never, Once, A few times, and Every day, scored as 0,1,2, and 3, respectively) and then the level of distress that the SE determines (scored from 1 to 10). Twenty questions refer to the prior week, while the last two questions (on changes in menstrual periods and weight gain) refer to the previous 3 months. The total scale score is given by the sum of the frequency of the items. The Italian translation of the scale is reported in Appendix A.

The clinician-rated UKU SE rating scale [12] is considered to be the gold standard for recording psychotropic drug–induced SE [18]. The original scale includes 48 items. The scale questions investigate the severity of the SE by defining specific discriminant parameters. The severity of the symptoms is defined as “no side effects” equal to 0, “mild side effects that do not interfere with the patient’s performance” equal to 1, “moderately” and “markedly” equal to 2 and 3, respectively. For the present study, we matched the GASS items with the UKU ones, adding items on nocturnal enuresis and breast pain that were not present in the original version of the UKU, as suggested by the manual. We replicated the procedure that Bock et al. applied in the Danish validation of the scale [18]. We adopted the same time frame for the GASS questions as the UKU ones. The total scale score was calculated by summing individual items matched to the GASS.

The WHO Disability Assessment Schedule (WHO-DAS) 2.0 is a generic self-rated tool used to measure health and disability levels in clinical practice [21]. There are two different versions of the instrument. The complete version contains 36 items, while the brief version—which we used for this study—includes 12 items, and it was also validated for patients with psychosis [22]. All the questions refer to the prior 30 days, asking for the level of difficulty in doing daily activities, ranging from “No difficulty,” equal to 1, to “Extreme or cannot do,” equal to 5. The sum of the items is proportional to the functional impairment.

The EuroQoL-5 dimensions-5 levels (EQ-5D-5L) is a quality-of-life screening tool [23]. It consists of two sections. The first part contains five Likert five-level questions regarding movement capacity, self-care, common activities, pain, and anxiety/depression. The second part consists of a visual analog scale (VAS) in which the patient should indicate his or her perceived health ranging from 0 to 100, where higher is better. We considered only the VAS for the present work, considering its more straightforward interpretation and correlation with other EQ-5D indices for patients with schizophrenia [24].

### 2.3. Translation and Validation Procedure

The translation procedure followed the prescriptions in the available literature [25]. First, we asked Prof. M. Taylor, the creator of the scale, for permission to translate it into Italian. Subsequently, the scale was translated from English into Italian by an Italian clinician proficient in the English language and an English, mother-tongue translator. With their consent and with the contribution of five patients, the two versions were then merged. Then, a clinician who was a native English speaker and proficient in the Italian language and an Italian mother-tongue translator back-translated the scale into English. The final version was then merged by consensus. Finally, all the documents were sent back to the GASS creator, who checked if the final version of the back-translated scale was in line with the original scale. After approval, the scale was administered to 10 patients, who confirmed the usability of the tool.

### 2.4. Raters

For the present study, three senior psychiatrists and three psychiatrists in training at the Psychiatry Unit administered the self-rated questionnaires. For the UKU SE rating scale administration, the senior and in-training psychiatrists performed a preliminary assessment on 10 patients to improve inter-rater reliability.

### 2.5. Statistical Analysis

#### 2.5.1. Structural Validity and Internal Consistency

The GASS is a multi-dimensional scale covering diverse SE. It was designed to measure the SE burden associated with AP drugs. Nevertheless, originally there were no subscales, and a total score could be utilized by summing the score of all items. Therefore, we conducted a confirmatory factor analysis (CFA) to determine the original one-factor construct of the scale. We used diagonally weighted least squares (DWLS) to estimate model parameters since the items are rated on an ordinal scale. The fit of CFA was examined with the chi-squared test, the comparative fit index (CFI; good fit when ≥0.95), the Tucker–Lewis index (TLI; good fit when TLI ≥0.95), and the root-mean-square error of approximation (RMSEA) and its 90% confidence intervals (CI; good fit when <0.06) [26]. We considered modifying the model by adding error covariances that could substantially improve the model*’*s fit when identified by modification indices. In addition, we examined internal consistency by calculating Cronbach’s α and its 95% CI (good internal consistency when ≥0.7) [27]. We also examined inter-item Spearman’s ρ correlations, and an average inter-item correlation between 0.2 and 0.4 indicated a good internal consistency [28].

#### 2.5.2. Concurrent Criterion Validity

We examined the agreement between the GASS and the UKU, which is regarded as the gold standard [18]. First, we paired the GASS items with the UKU clinician-administered items (Table A1) after dichotomizing both of them between present and absent. The sensitivity, specificity, positive predictive value (PPV), and negative predictive value (NPV) were calculated using the UKU items. Moreover, the phi coefficients of associations between dichotomized GASS and UKU items were calculated to obtain a good proxy of concurrent criterion validity between the scales [29]. Subsequently, the phi value was interpreted according to Rea and Parker*’*s anchor points [30]. We also investigated the relationship between the total scores of GASS and UKU with Spearman’s ρ (good agreement when ρ ≥ 0.7) [27].

#### 2.5.3. Hypothesis Testing for Construct Validity

We examined construct validity by investigating the relationship between the GASS total and functional impairment (measured with the WHO-DAS 2.0) or perceived quality of life (measured with the EQ-5D-5L VAS) by using Spearman’s ρ (good construct validity when |ρ| ≥ 0.5) [31]. We also investigated the relationship between frequency and distress scores of the individual GASS items using Spearman’s ρ. We examined differences in the GASS total score between patient subgroups (e.g., sex, diagnosis, employment status, etc.) using the Mann–Whitney U test or checking whether any correlation was found between GASS total score and demographic and illness-related variables.

#### 2.5.4. Clinical Feasibility

We recorded the timing of administering the instrument and the questions that the participants asked the clinicians when answering the GASS questionnaire.

The analyses were performed with caret [32], psych [33], and lavaan [34] packages with the RStudio IDE (integrated development environment) [35].

## 3. Results

### 3.1. Sample Characteristics

For the present study, we recruited 111 participants between September 2021 and April 2022. Due to missing frequency data needed for clinical validation, 11 participants were excluded from the analyses. In total, 100 participants provided all the data for the GASS frequency items, while only 81 completed the distress section. The sample comprised 69 patients with schizophrenia and 31 patients with bipolar spectrum disorders. All patients except four, who were taking typical AP only, were taking second-generation AP. 12 patients were treated with a combination of typical and atypical AP. Paliperidone, olanzapine, aripiprazole, and risperidone were the most prescribed (in 31%, 23%, 20%, and 12% of participants, respectively). Other AP were prescribed to less than 10 participants. A detailed description of the patients’ demographics and illness-related data is reported in Table 1.

### 3.2. Structural Validity and Internal Consistency 

#### 3.2.1. Structural Validity

The CFA did not find a good fit for the primary model of the one-factor construct of GASS (chi-squared = 309.81, degrees of freedom (df) = 189, *p*-value < 0.001; CFI = 0.81; TLI = 0.79; and RMSEA = 0.080, 90%CI [0.064, 0.096]). The five most influential modification indices were unique residual correlations concerning items 17 or 18 (items concerning “Gynecomastia/Breast pain“ and “ Galactorrhea“, which were both rare in our sample, Table 2) (Table A2). Therefore, we re-specified the model by adding these correlations, and the fit was better (chi-squared = 247.14, degrees of freedom (df) = 184, *p*-value = 0.001; CFI = 0.90; TLI = 0.89; and RMSEA = 0.059, 90%CI [0.038, 0.077]). Standardized loadings of the items on the total score are presented in Table A3.

#### 3.2.2. Internal Consistency

Cronbach’s α for the total score was 0.81, 95%CI [0.76, 0.86]. Inter-item correlations are presented in Figure A1, and the average inter-item correlation was 0.17. Medium to strong inter-item correlations (|ρ| > 0.4) were observed between items 6 “My hands or arms have been shaky” and 9 “My movement or walking have been slower than usual” (ρ = 0.51); items 2 “I felt drugged or like a zombie” and 12 “My mouth has been dry” (ρ = 0.48); items 12 and 19 “I had problems enjoying sex” (ρ = 0.47); items 4 “I felt my heart beating irregularly or unusually fast” and 6 (ρ = 0.47); items 5 “My muscles have been tense or jerky” and 11 “My vision has been blurry” (ρ = 0.45); items 4 and 5 (ρ = 0.44); items 3 “I felt dizzy when I stood up and/or have fainted” and 4 (ρ = 0.44); items 3 and 11 (ρ = 0.43); items 11 and 20 “I have had problems getting an erection” or 21 “I have noticed a change in my periods” (ρ = 0.41); items 7 “My legs have felt restless and/or I couldn’t sit still” and 10 “I have had uncontrollable movements of my face or body” (ρ = 0.41); items 11 and 13 “I have had difficulty passing urine” (ρ = 0.41); and items 5 and 6 (ρ = 0.40).

### 3.3. Criterion Validity

The sensitivity, specificity, PPV, and NPV are summarized in Table 2. Sensitivity ranged from 39.3% (“Asthenia”) to 99% (“Galactorrhea”), while specificity ranged between 20% (“Hyperkinesia”) and 100% (“Galactorrhea”). PPV ranged from 63.3% (“Nocturnal enuresis”) to 100% (“Galactorrhea”), while NPV ranged from 4.8 (“Hyperkinesia”) to 90% (“Nocturnal enuresis”). The phi correlation coefficient ranged between 0.01 (“Hyperkinesia”) to 0.70 (“Galactorrhea”). There was a strong or relatively strong association between GASS and UKU items for eleven items, a moderate association for six, and a weak or negligible association for the remaining ones.

There was a fair agreement between the GASS total and UKU total (ρ = 0.67, *p*-value < 0.001, Figure 1A).

### 3.4. Hypothesis Testing for Construct Validity

#### 3.4.1. Correlation between Functioning and Perceived Health

We found that a higher GASS total score was correlated with more functional impairment as indicated by higher scores on the WHO-DAS 2.0 (ρ = 0.45, *p* < 0.001, Figure 1B) and worse perceived health as indicated by lower scores on the VAS of 5Q-5D-5L (ρ = −0.4, *p* < 0.001, Figure 1C).

#### 3.4.2. Correlation between the Frequency of a Side-Effect and the Distress Caused to the Patient

The relationship between the frequency of a side-effect and the distress caused to the patient is presented in Table 3 (in 81 patients that rated distress). There were indications that some SE were more distressing when they were more frequent, e.g., sleepiness (item 1), parkinsonism, and dyskinesia (items 6, 9, and 10), and (anti)cholinergic side-effects (items 8, 12, and 13). In contrast, other SE may be equally distressing irrespective of their frequency, e.g., confusion or dizziness (items 2 and 3), akathisia (item 7), and sexual dysfunction (item 19).

#### 3.4.3. Differences between Subgroups

The GASS total score did not differ in any dichotomous comparison, except between patients providing distress data who had lower scores than those who did not complete the distress GASS column. The GASS total score did not correlate with any continuous variable (e.g., age, olanzapine equivalents dose, etc.).

### 3.5. Clinical Feasibility

As previously reported, 11 participants did not completely understand how to fill in the distress column. The median completion time for the GASS scale was 4:42 min. In Table 4, we report the most common questions asked by our participants.

## 4. Discussion

The present study aimed to translate the GASS into Italian and to validate it as a measure of the degree of the AP-SE burden.

In terms of structural validity, the one-factor definition of the scale might be inappropriate, requiring further investigation on the scale structure. However, it should be considered that the CFA statistics improved when considering the correlations of items 17 and 18 with some other ones. Those items could have impacted the one-factor analysis because of their rarity in our sample. Nevertheless, their paucity does not differ much from what was found in the CATIE study, which aimed to compare the effectiveness of conventional and atypical AP medications used for the treatment of schizophrenia [36].

For most of the individual SE investigated by the GASS against the UKU, the sensitivity and specificity in their identification were higher than 70%, resulting in a fair tool for AP-SE screening. On the other hand, even if the PPV of the GASS items was relatively high in most cases, the same cannot be said for the NPV. This means that patients may have been judged to have a specific SE during the clinical administration of the UKU but they did not recognize the same SE with the GASS. However, given that the primary aim of the GASS is to measure SE distress in patients, we consider our results acceptable for that aim.

The correlation between the GASS and UKU items ranged from negligible to strong [29,30]. In particular, it was strong and relatively strong for 11 items (orthostatic hypotension, palpitations, tremor, dry mouth, dysuria, nausea and vomiting, galactorrhea, sexual dysfunction, erectile dysfunction, menstruation changes, and weight gain). The correlation of the other items was lower. It is likely that the disproportion between the symptoms identified by the patient and the clinician is the result of the greater tendency of patients to report symptoms that determine distress [37] as present but are otherwise recognized as mild or absent by the clinician [38].

In line with a previous similar study [18], the items “asthenia” and “sedation“ did not perform well, but as opposed to previous findings, our results showed high specificity for the former. Unlike the other study, the GASS items investigating neurological SE showed fair sensitivity. It should be noted that the items exploring hypokinesis and hyperkinesis showed the former to be specific but not sensitive and the latter sensitive but not specific. This was probably conditioned by the fact that during the examination, the clinician tended to notice the global slowness of the patient more, which tends to be more evident as the disorder progresses [39], rather than hyperkinetic movements that show a fluctuating trend.

Regarding construct validity, the present work confirmed an inverse correlation between the total score of the GASS scale, functional impairment, and quality of life. The total score is proportional to the UKU total score in the items corresponding to the GASS scale. Moreover, we found that the frequency of SE is proportional to the distress that those SE determine. This leads to the consideration that the GASS might be preferentially used to estimate the SE burden rather than to identify the SE.

The translation and validation study of the GASS scale we performed adds to the studies already available [18,19,20]. As previously highlighted in the Greek and Arabic translations [19,20], the scale maintains good internal consistency, an element not investigated by the Danish validation [18]. However, consistency with the gold standard (measured by criterion validity) was not as satisfactory as in the Danish validation, although relatively robust for more than half of the items. Finally, the CFA, characterized by a different analytical approach to the one applied by Arab researchers [20], showed that the one-factor analysis did not fit well. Therefore, further exploratory factor analysis could be carried out to elucidate the scale structure.

The present study has some limitations. Firstly, it should be considered that the staff was trained before beginning the study using the UKU scale. We believe that this impacted on maintaining inter-rater reliability, considering that recruitment occurred within 7 months. This may have had an impact on criterion validity. Secondly, it should be considered that in order to be in line with Bock’s work [18], we used an instrument that was slightly modified from the one originally conceived by the creators of the GASS [17], considering the distress parameter to be continuous and not dichotomous. On the one hand, we could conduct more in-depth analyses; on the other hand, the distress column was considered ancillary in the original GASS and it is not used to calculate the total score of the scale. We did not find good reasons to maintain the continuous distress column, and we suggest using the original dichotomized form. Thirdly, given the small number of events found according to the GASS in terms of galactorrhea, menstrual cycle alterations, gynecomastia, and nocturnal enuresis, our sample size is unlikely to clarify sufficiently whether the scale is capable of adequately identifying them in clinical practice. Furthermore, in the present study, we did not measure either the discriminative ability of the tool or the test-retest. Only one of the other translations provided data on test-retest results, suggesting a good agreement between the administrations [19], similar to the original validation [17]. Additionally, it should be noted that the patients were all stable, most of them were treated with only second-generation antipsychotics, and they reported intermediate scores on the GASS scale on average. These sample characteristics may impact the generalizability of the present study. Finally, this study aimed to validate the Italian translation of the original scale. A sample size of at least 100 participants is suggested for this purpose. However, models with multiple parameters (such as when using the DWLS estimator) may need much larger samples (e.g., 20 participants for each parameter) [40].

The strengths of this research include the detailed analysis of the GASS, which integrates and expands the work of other researchers mentioned regarding the tool. This is relevant for future development of similar instruments. Moreover, the study involved some inpatients, not considered in previous studies, which extends the generalizability of the results. The validation followed the current standards for translating scales, maintaining the original face validity of the instrument. We provided a pragmatic instrument to measure AP-SE distress, requiring only 5 minutes to be completed, previously unavailable in a validated form for Italian patients with psychotic disorders.

## 5. Conclusions

The Italian translation and validation of the GASS adds a valuable tool to the patient-reported outcome measures (PROMs) that patients could benefit from. In addition, increased attention by clinicians to AP-SE may ensure an improvement in patients’ quality of life and psychosocial functioning. However, clinicians should integrate it with a clinical interview, as GASS appears more suitable as a screening tool than a diagnostic tool.

## Figures and Tables

**Figure 1 brainsci-12-00891-f001:**
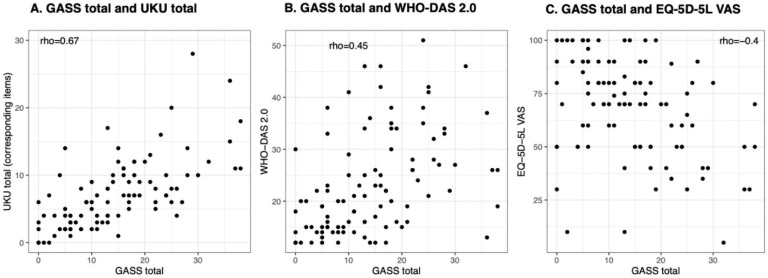
These figures illustrate the Spearman’s correlation coefficients between: (**A**) the Glasgow Antipsychotic Side-Effect Scale (GASS) total score and the Udvalg for Kliniske Undersøgelser (UKU) total score, where only the items corresponding to the GASS were considered for the computation; (**B**) the GASS total and the WHO Disability Assessment Schedule 2.0 (WHO-DAS 2.0); (**C**) the GASS and the EuroQoL-5 dimensions-5 levels Visual Analogue Scale (EQ-5D-5L VAS).

**Table 1 brainsci-12-00891-t001:** Demographics and illness-related data.

-		Total Sample (N = 100)	Schizophrenia Spectrum Disorders (N = 69)	Bipolar Spectrum Disorders (N = 31)
		Demographic Data		
	**Median age**	47 (IQR: 37–55)	45 (IQR: 36–53)	52 (IQR: 37–58)
**Sex ***	**Female**	39	21 (30%)	18 (58%)
	**Male**	61	48 (70%)	13 (42%)
**Education**	**Primary school**	10	9 (12%)	1 (3%)
	**Secondary school**	35	25 (36%)	10 (32%)
	**High school**	43	29 (42%)	14 (45%)
	**Higher education**	12	6 (9%)	6 (19%)
**Marital** **Status ***	**Unmarried**	72	54 (78%)	18 (58%)
	**Married**	28	15 (22%)	13 (42%)
**Employment status**	**Unemployed**	75	51 (74%)	24 (77%)
	**Employed**	25	18 (26%)	7 (23%)
**Smoking status**	**Non-smokers**	50	33 (48%)	17 (55%)
	**Smokers**	50	36 (52%)	14 (45%)
**Concomitant pathologies**	**Dysthyroidism**	9	7 (10%)	2 (6%)
	**Diabetes**	8	7 (10%)	1 (3%)
	**Hypertension**	14	8 (12%)	6 (19%)
	**Hyper-** **cholesterolemia**	7	5 (7%)	2 (6%)
		Illness-related data		
	**GASS total**	13 (IQR: 6–21)	13 (IQR: 6–21)	13 (IQR: 6–20)
	**UKU total ^+^**	6 (IQR: 3–10)	6 (IQR: 3–10)	6 (IQR: 4–10)
	**Median illness** **duration in years**	13 (IQR: 5–20)	11 (IQR: 5–20)	15 (IQR: 6–20)
	**Median lifetime** **hospitalizations**	2 (IQR: 0–3)	1 (IQR: 0–3)	2 (IQR: 1–4)
	**Median olanzapine oral equivalents in milligrams**	12.33 (IQR: 8–19)	13.3 (IQR: 8–20)	10 (IQR: 5–15)
**Antipsychotics administration route**	**Oral**	73	49 (71%)	24 (77.5%)
	**Injection**	15	10 (14.5%)	5 (16%)
	**Oral plus injection**	12	10 (14.5%)	2 (6.5%)
**Mono/polypharmacy**	**Monotherapy**	71	47 (68%)	24 (77%)
	**Polytherapy**	29	22 (32%)	7 (23%)
**Concomitant psychotropics**	**Mood stabilizers ***	49	24 (35%)	25 (81%)
	**Antidepressants**	36	21 (30%)	15 (48%)
	**Benzodiazepines ***	41	22 (32%)	19 (61%)
**Setting ***	**Outpatient**	90	65 (94%)	25 (81%)
	**Inpatient**	10	4 (6%)	6 (19%)

IQR: interquartile ranges; GASS: Glasgow Antipsychotic Side Effect Scale; UKU: Udvalg for Kliniske Undersøgelser (UKU) side-effects scale; * Schizophrenia and bipolar spectrum significantly differ for these variables; ^+^ only the items corresponding to the GASS were considered for the computation.

**Table 2 brainsci-12-00891-t002:** GASS and UKU scales, events, and validation measures (sensitivity, specificity, positive predictive value, negative predictive value, phi correlation coefficient).

Corresponding Items		Side-Effect Present		Concurrent Validity Measures					
GASS	UKU	GASS Events (%)	UKU Events (%)	Sens. (%)	Spec. (%)	PPV (%)	NPV (%)	Phi (ϕ)	ϕ Int.
1	Asthenia	69	44	39.3	79.5	71	50.7	0.20	M
2	Sedation	28	25	76	40	79.2	35.7	0.15	W
3	Orthostatic hypotension	27	21	82.3	61.9	89	48.1	0.41	RS
4	Palpitations	41	24	75	91.7	96.6	53.7	0.58	RS
5	Dystonia	34	8	68.5	62.5	95.5	14.7	0.18	W
6	Tremor	41	26	75.7	88.5	94.9	56.1	0.57	RS
7	Akathisia	29	18	79.3	66.7	91.5	41.4	0.39	M
8	Increased salivation	26	3	75.3	66.7	98.6	7.7	0.16	W
9	Hypokinesia	53	24	55.3	79.2	89.4	35.8	0.29	M
10	Hyperkinesia	21	5	78.9	20	94.9	4.8	0.01	N
11	Blurred vision	29	15	74.1	46.7	88.7	24.1	0.16	W
12	Dry mouth	53	42	65.5	78.6	80.9	62.3	0.44	RS
13	Dysuria	22	13	85.1	69.2	94.9	40.9	0.44	RS
14	Nausea/Vomiting	26	15	82.4	73.3	94.6	42.3	0.45	RS
15	Nocturnal enuresis	10	42	98.3	21.4	63.3	90	0.32	M
16	Polyuria/Polydipsia	53	33	58.2	75.8	83	47.2	0.32	M
17	Gynecomastia/Breast pain	9	9	93.4	33.3	93.4	33.3	0.27	M
18	Galactorrhea	2	1	99	100	100	50	0.70	S
19	Sexual dysfunction	30	49	88.2	49	64.3	80	0.41	RS
20	Erectile dysfunction	18	16	91.7	68.8	93.9	61.1	0.58	RS
21	Menstruation changes	4	6	98.9	50	96.9	75	0.59	RS
22	Weight gain	42	33	80.6	87.9	93.1	69	0.65	S

GASS: Glasgow Antipsychotic Side Effect Scale; UKU: Udvalg for Kliniske Undersøgelser side-effects scale; Sens.: sensitivity; Spec.: specificity; PPV: positive predictive value; NPV: negative predictive value; Int.: interpretation (based on Rea & Parker anchor points [30]).

**Table 3 brainsci-12-00891-t003:** Spearman’s correlation between frequency and distress of individual items of GASS.

Item	Score of 0 “Never”		Score of 1 “Once”		Score of 2 “A Few Times”		Score of 3 “Every Day”		Spearman’s ρ between Scores > 0 and Distress
	% pts	Distress Median (IQR)	% pts	Distress Median (IQR)	% pts	Distress Median (IQR)	% pts	Distress Median (IQR)	
1	33.3%	NA	11.1%	3 (1, 5)	38.3%	6 (3, 7)	17.3%	9.5 (8, 10)	**0.56, *p* = 0**
2	74.1%	NA.	6.2%	7 (5, 7)	16%	7 (5, 9)	3.7%	8 (8, 9)	0.28, *p* = 0.215
3	76.5%	NA.	4.9%	8.5 (6.8, 10)	14.8%	6 (4.2, 10)	3.7%	8 (4.5, 9)	−0.2, *p* = 0.411
4	65.4%	NA	6.2%	5 (5, 8)	22.2%	5 (2.2, 8)	6.2%	10 (9, 10)	0.36, *p* = 0.058
5	71.6%	NA.	6.2%	7 (7, 7)	18.5%	7 (5, 8.5)	3.7%	10 (10, 10)	0.3, *p* = 0.17
6	67.9%	NA.	6.2%	7 (2, 7)	18.5%	7 (4.5, 8.5)	7.4%	10 (9.2, 10)	**0.43, *p* = 0.029**
7	74.1%	NA	2.5%	6.5 (6.2, 6.8)	16%	6 (5, 8)	7.4%	8.5 (5.5, 10)	0.1, *p* = 0.671
8	80.2%	NA	4.9%	3.5 (2.5, 5)	8.6%	8 (5, 9)	6.2%	10 (9, 10)	**0.63, *p* = 0.009**
9	49.4%	NA	8.6%	6 (2.5, 7.5)	28.4%	7 (4.5, 8)	13.6%	10 (8, 10)	**0.45, *p* = 0.003**
10	85.2%	NA	7.4%	5 (3.5, 6.5)	7.4%	9 (6.5, 10)	0%	NA	**0.65, *p* = 0.022**
11	76.5%	NA	1.2%	5 (5, 5)	17.3%	7 (4.2, 9)	4.9%	10 (9, 10)	0.39, *p* = 0.095
12	50.6%	NA	7.4%	1 (1, 3.2)	27.2%	5 (3.2, 7.8)	14.8%	10 (8.8, 10)	**0.65, *p* = 0**
13	82.7%	NA	4.9%	3 (1.8, 4.8)	8.6%	8 (6, 8)	3.7%	9 (8, 9.5)	**0.62, *p* =0.018**
14	80.2%	NA	1.2%	5 (5, 5)	16%	6 (5, 8)	2.5%	10 (10, 10)	0.47, *p* = 0.065
15	91.4%	NA	1.2%	1 (1, 1)	6.2%	9 (6, 10)	1.2%	10 (10, 10)	0.63, *p* = 0.133
16	50.6%	NA	8.6%	2 (1, 6)	22.2%	5 (4.2, 6)	18.5%	7 (4.5, 10)	**0.33, *p* = 0.036**
17	91.4%	NA	1.2%	4 (4, 4)	6.2%	1 (1, 4)	1.2%	10 (10, 10)	0.45, *p* = 0.306
18	98.8%	NA	0%	NA	0%	NA	1.2%	10 (10, 10)	NA
19	69.1%	NA	2.5%	7.5 (7.2, 7.8)	16%	5 (1, 6)	12.3%	6.5 (2, 10)	0.04, *p* = 0.863
20	71.7%	NA	3.5%	1 (1, 1)	22.9%	8.5 (6.2, 10)	22.9%	10 (10, 10)	0.44, *p* = 0.137
21	88.6%	NA	NA	NA	NA	NA	26.4%	8.5 (6.2, 9.2)	NA
22	56.8%	NA	NA	NA	NA	NA	43.2%	7 (5, 9)	NA

pts: patients; IQR: interquartile range; NA: not available.

**Table 4 brainsci-12-00891-t004:** Participants’ questions regarding the completion of the GASS.

Question related to GASS item 12, dry mouth, (n = 2)
What does “dry mouth” (*bocca secca*, in Italian) mean?
Question related to GASS item 22, weight gain, (n = 3)
Is it asking for gain or loss of weight? (*preso* or *perso,* respectively, in Italian)
Question related to GASS item 20, erection, (n = 3):
What does “Men Only” mean?
Questions related to GASS Distress column (n = 9):
Is it a tick box? What does the number mean? If the main answer is “Never”, which “distress” might be chosen?
Questions related to general GASS form, (n = 2):
What is the difference between “once” and “almost never”? What if it is neither “once” nor “a few times”?
Other Questions, (n = 4):
Question related to GASS item 1, day sleepiness (n = 1) What does “sleepy” (*assonnato*, in Italian) mean? Question related to GASS item 2, asthenia (n = 1) What does “like a zombie” (*come uno zombie*, in Italian) mean? Question related to GASS item 5 (n = 1) What does “jerky” (*sono andati incontro a contrazioni*, in Italian) mean? Question related to GASS item 18 (n = 1) How can fluid come from nipples?

## Data Availability

The data presented in this study are available on reasonable request from the corresponding author.

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
