# Peer review of "Validation of the Glasgow Antipsychotic Side-Effect Scale (GASS) in an Italian Sample of Patients with Stable Schizophrenia and Bipolar Spectrum Disorders"

_brainsci, 2022, doi:10.3390/brainsci12070891_

Round 1
Reviewer 1 Report
The paper of Rodolico and colleagues aimed to translate Glasgow Antipsychotic Side-Effect Scale (GASS) to Italian as well as validate it against Udvalg for Kliniske Undersøgelser (UKU). While scientific soundness of study relatively low the study accurately performed and the manuscript well wrote. With some methodological exceptions the work of Rodolico et al., in general replicates the study of Schouby Bock et al., (2020) performed on Danish patients. The one significant difference from Schouby Bock’s study is implication of patients with bipolar disorder. The fact that dosage of antipsychotic prescribed for schizophrenia and bipolar patients usually differ tree to ten times may one of the limitations in the present study. It is not clear what antipsychotics were used by patients included in the study and what were they dosages. So, I think that authors should at least discuss this moment in the paper.
Author Response
Q1. The paper of Rodolico and colleagues aimed to translate Glasgow Antipsychotic Side-Effect Scale (GASS) to Italian as well as validate it against Udvalg for Kliniske Undersøgelser (UKU). While scientific soundness of study relatively low the study accurately performed and the manuscript well wrote. With some methodological exceptions the work of Rodolico et al., in general replicates the study of Schouby Bock et al., (2020) performed on Danish patients. The one significant difference from Schouby Bock’s study is implication of patients with bipolar disorder.
R1. Thank you very much for your positive feedback and the constructive suggestions.
Q2: The fact that dosage of antipsychotic prescribed for schizophrenia and bipolar patients usually differ tree to ten times may one of the limitations in the present study.
R2: We appreciate the comment of the reviewer. We acknowledge that, as suggested by the reviewer, the dosage of antipsychotic prescribed may vary between patients. However, in our sample. the average antipsychotic dosage did not differ between schizophrenia and bipolar spectrum patients. Moreover, the comparison of GASS scores between groups did not show statistically significant differences.
Q3: It is not clear what antipsychotics were used by patients included in the study and what were they dosages. So, I think that authors should at least discuss this moment in the paper.
R3: We thank the reviewer for the suggestion. We did not report all the antipsychotics used in the study. To avoid redundancy and overreporting of uninformative data concerning only a minority of participants, we decided to report only the higher frequency data in full. Specifically, in the results section, the reader can find the following text:
All patients except 4, who were taking typical AP only, were taking second-generation AP. 12 patients were treated with a combination of typical and atypical AP. Paliperidone, olanzapine, aripiprazole, and risperidone were the most prescribed (in 31%, 23%, 20%, and 12% of participants, respectively). Other AP were prescribed to less than 10 participants.
Moreover, in Table 1, dosages are reported as olanzapine equivalents. The reason for showing data this way prevented scattered reporting. In addition, it enabled us to report a single dosage for those patients taking a combination of antipsychotics that the equivalence allowed to sum up.
Reviewer 2 Report
-minor suggestion: Perhaps describing why the GASS is now a consensus would be useful including a brief description of the scale (number of quesitons, scoring, etc). I undestand some of this information is in the methods but might be useful in the introduction
-Did you have a wide variety of 1st and 2nd generation antipsychotics to increase external validity of your scale in Italian?
-What was the aimed recruitment for this study or just convenience sampling? Did you have an estimated number to reach 80% power? I see you discuss a suggested sample size of 100 in your limitations discussion, is the recommendation because it achieves a given power?
-for the original, English development of the GASS, did the research suggest that is was mostly useful as a screening rather than diagnostic tool?
Author Response
Q1. minor suggestion: Perhaps describing why the GASS is now a consensus would be useful including a brief description of the scale (number of questions, scoring, etc). I undestand some of this information is in the methods but might be useful in the introduction
R1. We shared the point of the reviewer and extended the paragraph in the introduction (new text highlighted):
The Glasgow Antipsychotic Rating Scale (GASS) was suggested by a consensus of over 100 professionals and users as the standard patient-reported outcome measure used to collect data about AP-SE[15]. The GASS scale selection process was based on a sequence of steps. Initially, two patents focus groups selected the most important outcomes for them. Subsequently, the stakeholders chose only the “essential” outcomes. Next, they assessed them with the COnsensus-based Standards for the selection of health outcomes Measurement Instruments (COSMIN) checklist to check the psychometric properties of measures, including reliability, validity, and responsiveness[16]. Finally, the GASS was chosen to monitor treatment side effects. It consists of a set of twenty-two self-explanatory questions expressed in easy-to-understand English. The patient indicates the frequency and the distress determined by the specific SE. The higher the sum of the frequencies of the SEs, the higher the burden the SEs determine. Its brevity permits the completion of the patient self-assessment in about five minutes[17].
Q2. Did you have a wide variety of 1st and 2nd generation antipsychotics to increase external validity of your scale in Italian?
R2. Most of the patients were taking second-generation antipsychotics. We acknowledge that this is a limitation of the present study. therefore, we have added the following sentence accordingly (new text highlighted):
Also, it should be noted that the patients were all stable, most of them were treated with only second-generation antipsychotics, and they reported intermediate scores on the GASS scale on average. These sample characteristics may impact the generalizability of the present study.
Q3. What was the aimed recruitment for this study or just convenience sampling? Did you have an estimated number to reach 80% power? I see you discuss a suggested sample size of 100 in your limitations discussion, is the recommendation because it achieves a given power?
R3. There are multiple references suggesting the optimal participants' number, but there is no consensus. We referred, as mentioned in the discussion, to Kline 2015. This number is also reported as sufficient in Terwee 2007, where the authors provide a set of quality criteria for health status questionnaires. The authors suggest that a “minimum number of 100 subjects” is acceptable for scale validation. Moreover, our study, along with AlRuthia 2018, has the highest numerosity among all the other validation or translation studies. In our study, we did not speculate too much on the scale structure, aware of the risk of underpowered data analyses.
Q4. for the original, English development of the GASS, did the research suggest that is was mostly useful as a screening rather than diagnostic tool?
R4. The authors of the original version of the instrument report the following sentence in their manuscript: “We aimed to construct and assess a new self-rating scale to detect the side effects of second-generation antipsychotics. This scale was designed to allow a timely, sensitive and reliable method of gathering information on the number and severity of side effects an individual suffers from.” So then, it seems it was built as a diagnostic tool. However, the diagnosis process implies establishing the presence/absence of a specific condition with higher accuracy than a screening tool. This could be achieved by clinician evaluation and not by using another screening instrument, such as the LUNSER scale. Therefore, similarly to Schouby Bock, we tried to establish if the tool could be considered “diagnostic” and we do not feel comfortable providing this strong suggestion based on our results.
References
- AlRuthia, Y., Alkofide, H., Alosaimi, F. D., Alkadi, H., Alnasser, A., Aldahash, A., Basalamah, A., & Alarfaj, M. (2018). Translation and cultural adaptation of Glasgow Antipsychotic Side-effects Scale (GASS) in Arabic. PloS one, 13(8), e0201225. https://doi.org/10.1371/journal.pone.0201225
- Kline RB. Principles and Practice of Structural Equation Modeling, Fourth Edition: Guilford Publications; 2015.
- Mokkink, L. B., Prinsen, C. A., Bouter, L. M., Vet, H. C., & Terwee, C. B. (2016). The COnsensus-based Standards for the selection of health Measurement INstruments (COSMIN) and how to select an outcome measurement instrument. Brazilian journal of physical therapy, 20(2), 105–113. https://doi.org/10.1590/bjpt-rbf.2014.0143
- Schouby Bock, M., Nørgaard Van Achter, O., Dines, D., Simonsen Speed, M., Correll, C. U., Mors, O., Østergaard, S. D., & Kølbæk, P. (2020). Clinical validation of the self-reported Glasgow Antipsychotic Side-effect Scale using the clinician-rated UKU side-effect scale as gold standard reference. Journal of psychopharmacology (Oxford, England), 34(8), 820–828. https://doi.org/10.1177/0269881120916122
- Terwee, C. B., Bot, S. D., de Boer, M. R., van der Windt, D. A., Knol, D. L., Dekker, J., Bouter, L. M., & de Vet, H. C. (2007). Quality criteria were proposed for measurement properties of health status questionnaires. Journal of clinical epidemiology, 60(1), 34–42. https://doi.org/10.1016/j.jclinepi.2006.03.012